# Interface-mediated hygroelectric generator with an output voltage approaching 1.5 volts

Yaxin Huang[1], Huhu Cheng[1,2], Ce Yang[1], Panpan Zhang[1], Qihua Liao[1], Houze Yao[1,2], Gaoquan Shi[2] & Liangti Qu [1,2,3]

Hygroelectricity is proposed as a means to produce electric power from air by absorbing gaseous or vaporous water molecules, which are ubiquitous in the atmosphere. Here, using a synergy between a hygroscopic bulk graphene oxide with a heterogeneous structure and interface mediation between electrodes/materials with Schottky junctions, we develop a high-performance hygroelectric generator unit with an output voltage approaching 1.5 V. High voltage (e.g., 18 V with 15 units) can be easily reached by simply scaling up the number of hygroelectric generator units in series, enough to drive commercial electronic devices. This work provides insight for the design and development of hygroelectric generators that may promote the efficient conversion of potential energy in the environmental atmosphere to electricity for practical applications.

[1] Key Laboratory for Advanced Materials Processing Technology, Ministry of Education of China, State Key Laboratory of Tribology, Department of Mechanical Engineering, Tsinghua University, Beijing 100084, PR China. [2] Department of Chemistry, Tsinghua University, Beijing 100084, PR China. [3] Beijing Key Laboratory of Photoelectronic/Electrophotonic Conversion Materials, School of Chemistry and Chemical Engineering, Beijing Institute of Technology, Beijing 100081, PR China. Deceased: Gaoquan Shi. Correspondence and requests for materials should be addressed to H.C. (email: huhucheng@tsinghua.edu.cn) or to L.Q. (email: lqu@bit.edu.cn)

The development of new energy conversion technologies is sought after to satisfy the enormous demands for electric power in our daily lives, from electronics to vehicles. Abundant energy exists in available sources such as sunlight, wind, and water and can be used to produce electric power using solar cells, wind turbines, hydroturbines, etc. Water-enabled energy harvesting systems arouse interest, since ~ 71% of the surface of the Earth is covered by water. In this regard, researchers have devoted great efforts to generate electricity by employing streaming potential[1], dragging potential[2], and waving potential[3] based on interactions between water and functional materials, relying on the interplay between the two. Hygroelectricity, a type of energy-harvesting concept for tapping electric power from potential energy of spontaneous water molecules in air, is of interest due to copious water in the atmosphere[4,5]. Recently, by using hygroscopic graphene oxide (GO), we have developed a series of hygroelectric generators (HEGs) that can directly produce electricity by absorbing abundant water molecules in air, showing excellent performance and environmental compatibility[6–14].

Generally, a typical HEG is composed of a pair of electrodes and hygroscopic materials with a chemical-gradient structure (Fig. 1a)[6,15]. When a HEG is exposed to air enriched with water molecules, spontaneous absorption and subsequent hydration of the hygroscopic materials result in plenty of free-charge carriers (e.g., protons) by an ionization effect[16,17]. Then, the directional diffusion of charge carriers under the function of a gradient structure in the hygroscopic materials gives rise to the electric potential between two electrodes that induce a current flow in external circuit, thus outputting electric power. The output voltage of developed HEGs has reached a considerable level (from 0.035 V to 0.7 V)[6,7,11,13] by the optimization of inner gradient materials of a HEG with a variety of strategies, such as chemical modification[7,8,13] and microstructure control[9,10]. However, the generated voltage is still too low to meet the demand of commercial electronics (>1 V).

Here, by adopting an effective synergy strategy of heterogeneous reconfiguration of oxygen-containing functional groups on hygroscopic GO and mediation of electrodes/materials interfaces with well-designed Schottky junctions (Fig. 1b), we develop a high-voltage HEG with output approaching 1.5 V. As shown in

Fig. 1b, the heterogeneous graphene oxide (h-GO) has an integrated structure with a thick GO layer under the gradient-reduced GO (grGO) layer. Compared with previous devices composed of single grGO, additional GO layers in the h-GO could supply more movable ions for feeding the vacancies within grGO induced by the directional migration of protons influenced by the gradient structure once hydrated with environmental water molecules in high humidity. On the other hand, a well-matched space charge zone is formed by the regulation of the interface between h-GO and electrodes (a pair of Ag/Au electrodes) with rational work function, which could efficiently tailor the flow of free electrons in an external circuit to be consistent with the direction of proton migration in h-GO (Fig. 1b). Thus, in comparison with previous devices, the output performance of this HEG reaches a high value of ~1.5 V. Moreover, a voltage of 18 V is achieved by simple connection of 15 units in series, easily powering commercial electronic devices.

## Results

**Fabrication and characterization of a hygroelectric generator.** First, directionally controlled laser irradiation is employed on GO bulk for reconfiguration of inner oxygen-containing functional groups (Fig. 2a and Supplementary Figs 1, 2). Unlike the complete reduction of the whole GO bulk by other methods[16,18–21], the directional laser irradiation is gradually weakened along with the depth increase of GO bulk[22–24]. Thus, the as-prepared h-GO has a thick unreduced GO layer under a thin grGO layer as the visible color difference indicated in the cross-sectional photo (inset Fig. 2b). X-ray photoelectron spectroscopy (XPS) demonstrates that the C/O atomic ratio for grGO on the top surface of h-GO is about 3:1, which is much higher than the value of its bottom GO side (2:1) (Supplementary Figs 3, 4). Energy-dispersive X-ray spectroscopy (EDS) spectra further indicate that the C/O atomic ratio decreases gradually within a height range of ~ 40 μm of grGO from the top to the bottom side, while the C/O atomic ratio almost remains unchanged in the unreduced GO section of h-GO (Fig. 2c). Moreover, the interlayer spacing of h-GO increases from 6.7 Å (grGO) to about 7.6 Å (GO), verified by X-ray diffraction (XRD) patterns (Fig. 2d). All above results reconfirm the successful construction of the unique heterogeneous structure of h-GO.

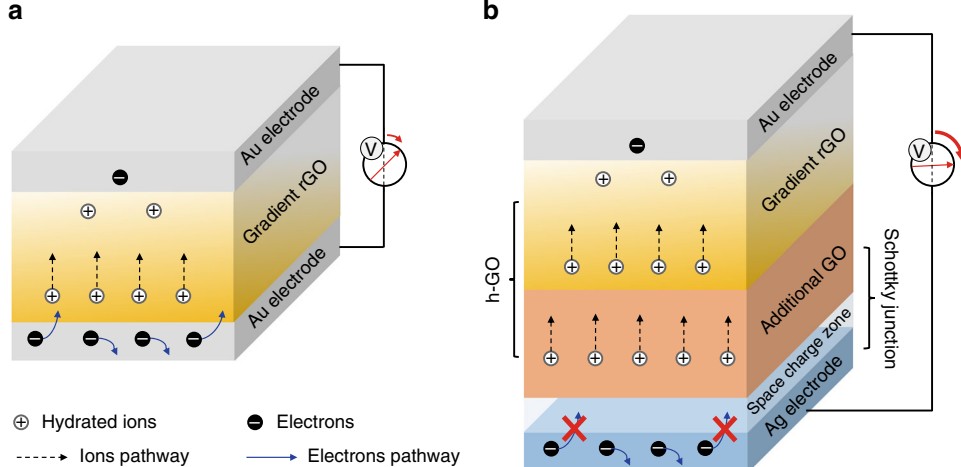

**Fig. 1** Additional feed-layer enhanced hygroelectric generator. Schematic of previous hygroelectric generator (HEG) composed of single gradient reduced graphene oxide (rGO) (**a**) and well-designed HEG (**b**) developed with heterogeneous graphene oxide (h-GO) and well-matched electrodes. **a** Movable hydrated ions dissociate in gradient rGO with concentration difference and diffuse from high-concentration side to low-concentration side. Induced free electrons in electrodes flow through both the external circuit and inner gradient rGO layer to balance the built-in electric potential. **b** Feed-layer with additional graphene oxide (GO) supplies the depletion of ions in gradient rGO, and special electrode/h-GO interface can form a well-matched space charge zone, which acts as a "gate" to block free electrons through h-GO and regulates its flow direction, largely promoting the electric output

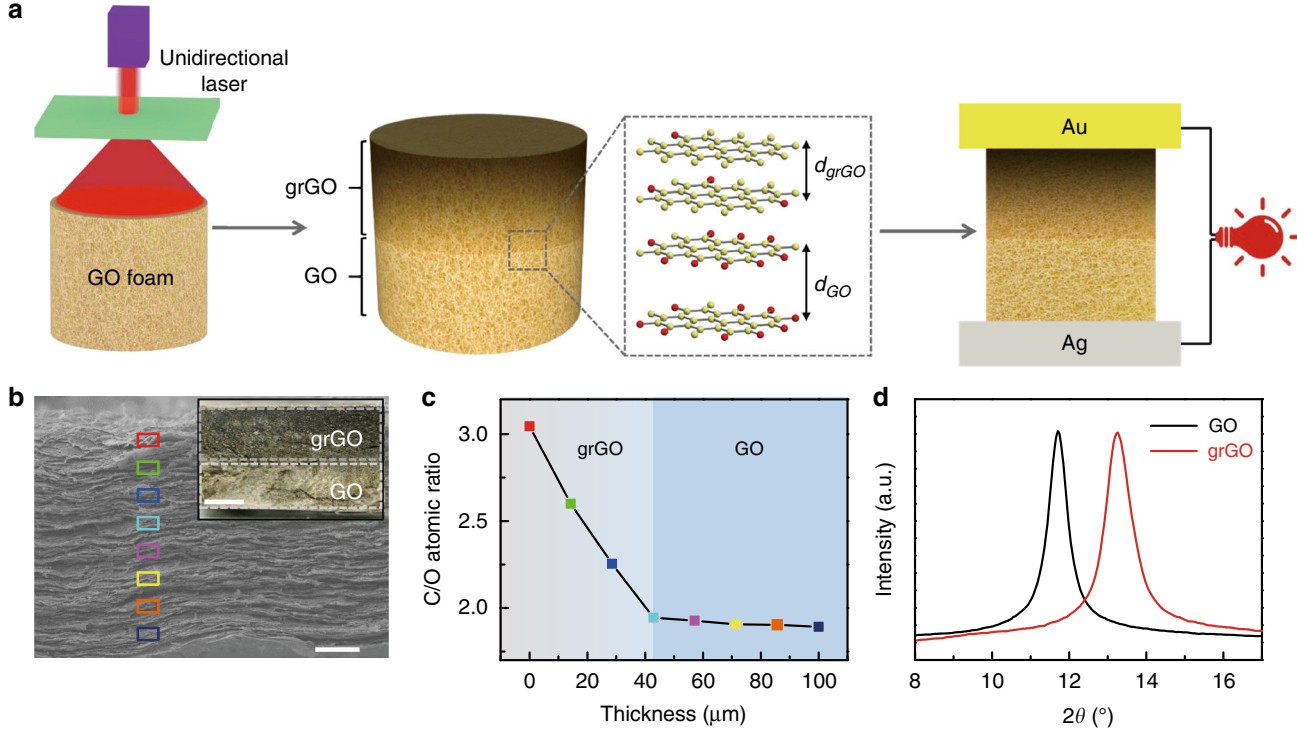

**Fig. 2** Preparation and characterization of heterogeneous graphene oxide. (**a**) Schematic illustration of the preparation process of heterogeneous graphene oxide (h-GO)-based hygroelectric generator (HEG). h-GO is composed of gradient reduced graphene oxide (grGO) with compact interlayer spacing and graphene oxide (GO). (**b**) Cross-sectional scanning electron microscopy (SEM) image of h-GO. Inset b is the sectional-viewed optical image of h-GO. Scale bars: **b**, 20 μm; inset, 0.2 mm. (**c**) Energy-dispersive spectroscopy (EDS) along the transversal direction of h-GO indicates a chemical C/O ratio gradient within the grGO, while a nearly constant C/O ratio through the GO region. (**d**) X-ray diffraction (XRD) pattern of top side (grGO) and bottom side (GO) of h-GO

**Electrical output performance of hygroelectric generator**. Benefiting from the heterogeneous distribution of oxygen-containing functional groups when h-GO is sandwiched between a pair of Ag and Au electrodes, one HEG (Ag/GO–grGO/Au) unit in which an Ag electrode is contacted with GO and an Au electrode is contacted with grGO produces an unprecedented high voltage of 1.25–1.52 V by absorbing water from an atmosphere with high relative humidity (RH) (variation in RH ($\Delta$RH) of 80% at 25 °C, Supplementary Figs 5, 6). The voltage is higher than that produced by electricity generators that employ gaseous and even liquid water[1–3,7,13,25–28]. The electrical signal raises up quickly within 2 s and gradually descends for about 600 s to its initial state (Fig. 3b) with RH evolution (10%–90%–10%), which is consistent with the hydration and dehydration speed of h-GO (Supplementary Fig. 7), indicating that electricity generation is originally induced by the water absorption from humid air. Meanwhile, a considerable short-circuit current of 98–136 nA is generated and the power density reaches up to 32 mW cm$^{-3}$ at the optimal resistance load of 10 MΩ (Fig. 3e, f). Meanwhile, this HEG (Ag/GO–grGO/Au) shows good stability after 50 cycling tests (Supplementary Fig. 8), further indicating its excellent electric generation ability by absorbing water molecular from humidity air.

**Working mechanism**. In the electricity generation process, positively charged protons could spontaneously diffuse from the GO side to the grGO side due to the hydration effect between h-GO and absorbed water molecules[6,7]. In addition, the suitable electrodes configuration induces a Schottky barrier at the electrodes/h–GO interface that would match well with the direction of diffusion of positively charged protons in inner

h-GO, greatly enhancing the final voltage output. In the HEG system (Ag/GO–grGO/Au) shown in Fig. 4a, Ag and GO has distinct work function that is about 4.26 eV and 4.9 eV[29], respectively, which demonstrates a prominent Schottky diode characteristic at the Ag/GO interface[30–32]. While the grGO/Au shows a typical Ohmic contact due to the high work function of Au (5.1 eV) than grGO (4.4–4.7 eV)[29,33,34], I–V curve on HEG (Ag/GO–grGO/Au) further confirms the Schottky junction characteristic of this device. On the other hand, a space charge zone will form at the Ag/GO interface, which would hold back the flow of negatively charged electrons from the Ag electrode to the GO layer, thus promoting charge separation in the whole h-GO. In contrast, when alternating the electrodes to construct HEGs with a configuration of Au/GO–grGO/Au and Ag/GO–grGO/Ag, the final voltage output decreases to 0.82 V and 0.8 V (Fig. 4d), respectively. The HEG (Au/GO–grGO/Au) lacks Schottky junctions, in which the flow of electrons from electrodes to h-GO is short of management, resulting in recombination of the positive and negative charges (Fig. 4b). The HEG (Ag/GO–grGO/Ag) has two opposite Schottky junctions that seriously weaken the diffusion of charges (Fig. 4c). Moreover, the same output sequence also shows in the case of current (Fig. 4e). Therefore, the rational synergy strategy of unique asymmetrical structure of h-GO and well-mediated electrode/material interface within device leads to the highly electric performance of HEG (Ag/GO–grGO/Au) in this study (Supplementary Figs 9, 10).

Above all, the directional migration of protons in h-GO and management on free electrons in the circuit are speculated to be beneficial to the charge partition in HEG. Thus, the proposed operation principle of as-designed HEG (Ag/GO–grGO/Au)

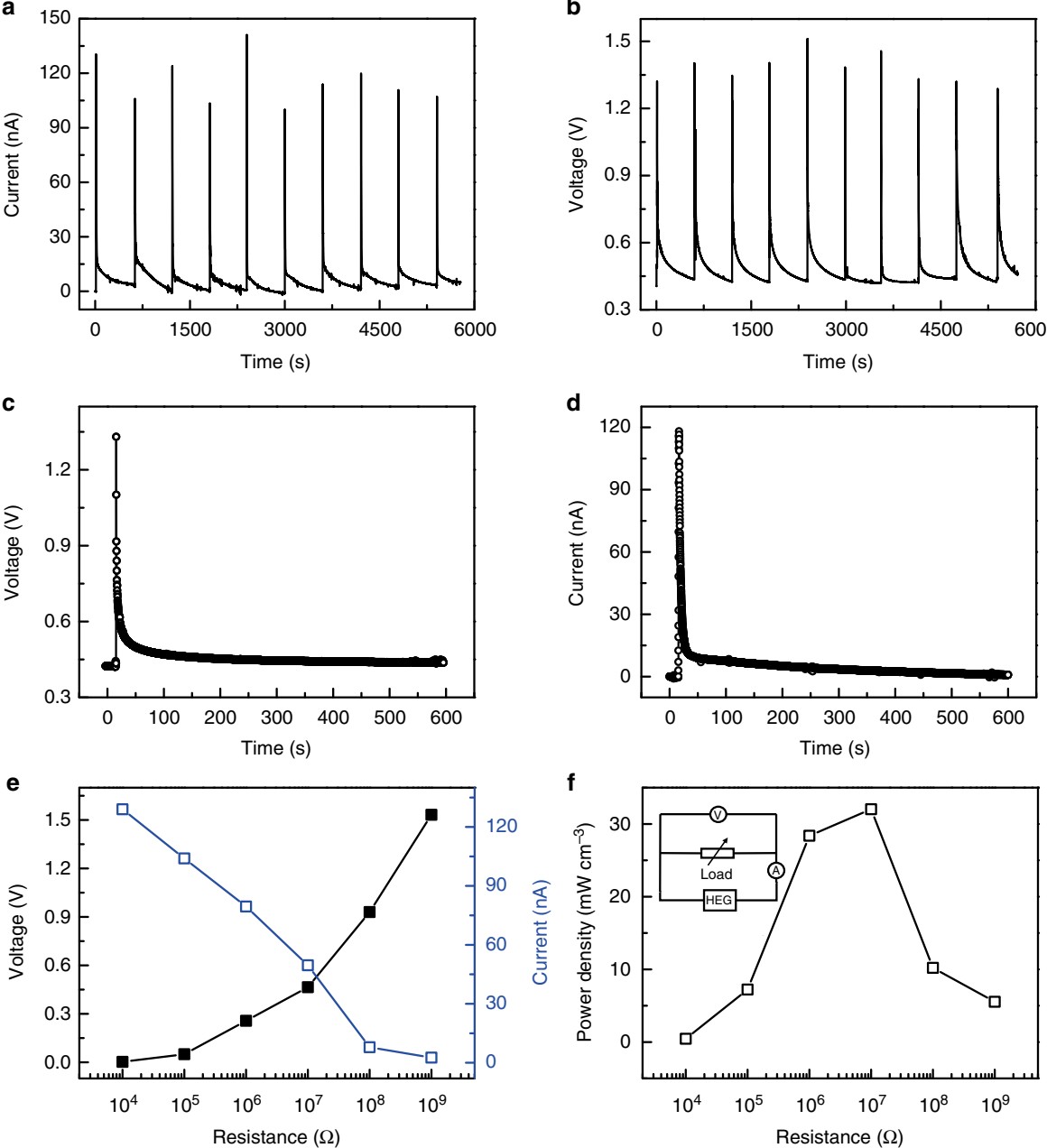

**Fig. 3** Electric output performance of hygroelectric generator. **a**, **b** Voltage and current output cycles of HEG (Ag/graphene oxide (GO)–gradient reduced graphene oxide (grGO)/Au) in response to the intermittent and periodic relative humidity variation (△RH = 80%). HEG (Ag/GO–grGO/Au) is composed of a pair of Au, Ag electrodes, and heterogenous graphene oxide (h-GO) in which the Au electrode contacts with graphene oxide (GO) side of h-GO, while Ag electrode contacts with gradient reduced graphene oxide (grGO) side of h-GO. **c**, **d** A single and representative cycle voltage and current output of HEG (Ag/GO–grGO/Au). **e**, **f** Output voltage (black curve, **e**), current (blue curve, **e**), and power density (**f**) of the HEG (Ag/GO–grGO/Au) with different electric resistors. The inset is an equivalent circuit diagram

could be depicted schematically in Fig. 5a. The two sections of h-GO, with thickness of $d_1$ (GO) and $d_2$ (grGO) and the relative dielectric constants $\varepsilon_1$ and $\varepsilon_2$, respectively, are sandwiched with a pair of asymmetrical Au and Ag electrodes. At a given RH atmosphere, the positively charged protons induce charges of $Q_1$, $Q_2$, and $Q_3$ at Au/grGO, GO/grGO, and GO/Ag interfaces, respectively. The electric field built by charge partition can be illustrated using a plate capacitance model, since the area size ($S$) of h-GO is several orders of magnitude larger than its thickness $(d_1 + d_2)$ in the present experiment[35]. From the Gauss model, the induced electric field strength at each section

is determined by

$$\text{Inside grGO} : E_1 = \frac{Q_2 - Q_1}{S\varepsilon_0\varepsilon_1} \quad (1)$$

$$\text{Inside GO} : E_2 = \frac{Q_3 - Q_2}{S\varepsilon_0\varepsilon_2} \quad (2)$$

where $\varepsilon_0$ is the vacuum permittivity. The directional migration of protons and charge separation firstly occurs in grGO from high to low oxygen-containing functional groups, then resulting

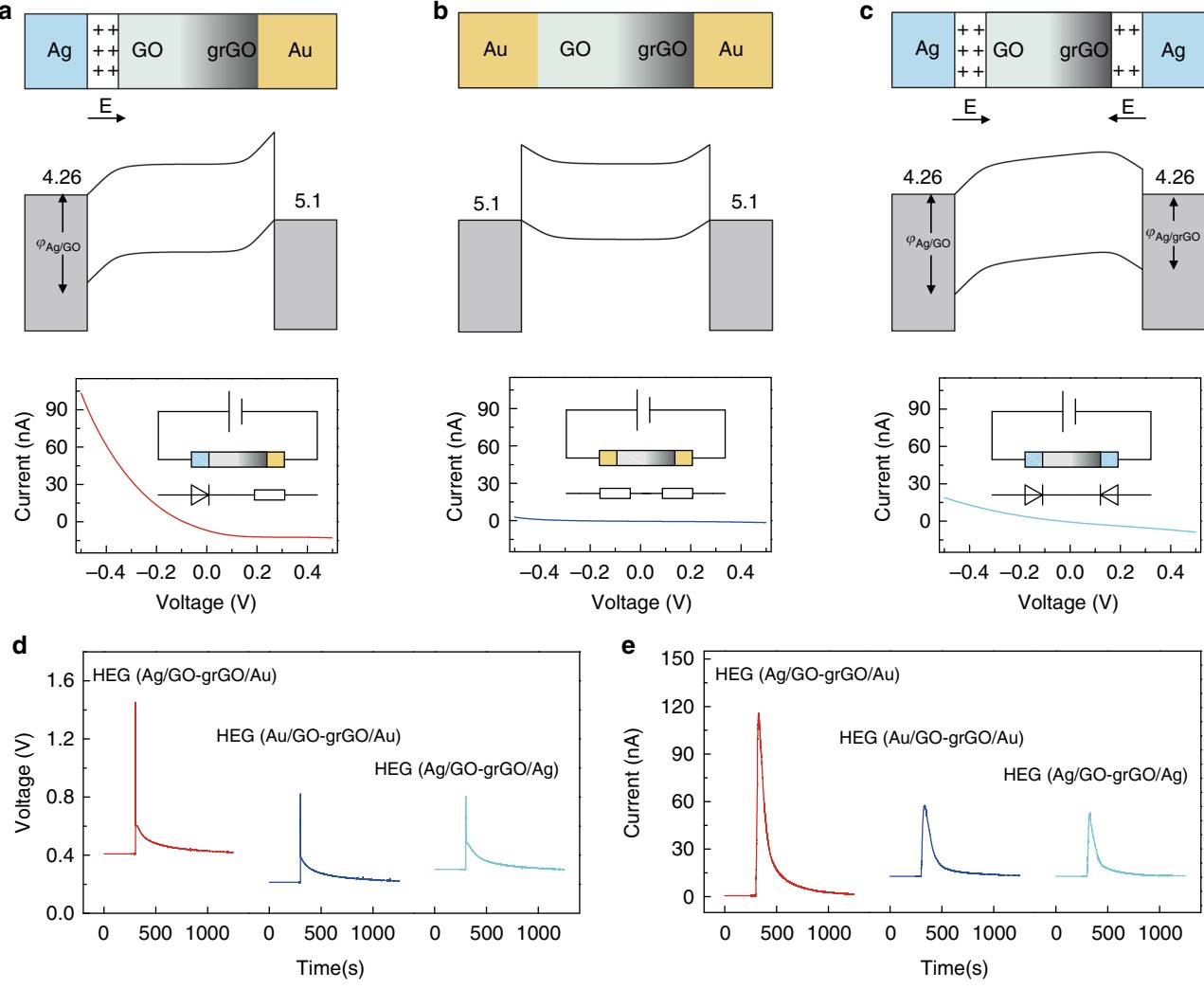

**Fig. 4** Comparison of hygroelectric generators with different configurations. **a–c** Device structure (top), energy band diagram (middle), and corresponding current–voltage characteristic curve (bottom) of (**a**) HEG (Ag/GO–grGO/Au) with a pair of Ag and Au electrodes, (**b**) HEG (Au/GO–grGO/Au) with two identical Au electrodes and (**c**) HEG (Ag/GO–grGO/Ag) with two identical Ag electrodes. The sandwiched heterogeneous graphene oxide (h-GO) between two metal electrodes is composed of graphene oxide (GO) and gradient reduced graphene oxide (grGO) parts. $\varphi$ indicates the barrier potential difference at the interface. Current–voltage curves show typical Schottky diode characteristic of HEG (Ag/GO-grGO/Au), Ohmic contact of HEG (Au/GO–grGO/Au) and two opposite Schottky contact of HEG (Ag/GO-grGO/Ag). The inset of current–voltage curve is the corresponding test circuit and equivalent circuit diagram. **d**, **e** Voltage and current output of above three hygroelectric generators with different device configuration

in the proton diffusion from GO to grGO and charge separation at GO region. Thus, the generated voltage ($V$) of the whole device can be given by

$$V = E_1 d_1 + E_2 d_2 + V_s \qquad (3)$$

where $V_s$ represents the Schottky barrier between GO and Ag electrode. Finally, we can obtain

$$V = \left( \frac{(Q_2 - Q_1)d_1}{S\varepsilon_0\varepsilon_1} + \frac{(Q_3 - Q_2)d_2}{S\varepsilon_0\varepsilon_2} + V_s \right) \qquad (4)$$

Equation (4) means the voltage ($V$) is proportional to the difference of charge quantity ($Q_2$–$Q_1$) and ($Q_3$–$Q_2$), and inversely proportional to relative dielectric constants of GO ($\varepsilon_2$) and grGO ($\varepsilon_1$). Firstly, the average surface potential of a single GO and grGO sheet is about $-2$ mV and $-27$ mV (EFM results in Fig. 5b, c and Supplementary Fig. 11), respectively, indicating an extremely remarkable difference of charge amount of h-GO

bulk[36,37]. Secondly, with the increase of RH, the charge amount of GO and grGO sheets presents an apparent increase as shown in Fig. S9, which couples with the decrease of relative dielectric constants of GO ($\varepsilon_2$) and grGO ($\varepsilon_1$)[38–41], accounting for the great enhancement of output voltage according to Equation (4). Thirdly, the Schottky barrier ($V_s$) between GO and Ag electrode could be slightly enhanced ascribed to the increased work function of GO with rising RH[41,42]. Finally, compared with previous devices that mainly relay on the first part $\left( \frac{(Q_2-Q_1)d_1}{S\varepsilon_0\varepsilon_1} \right)$ of the speculated Equation (4), this HEG (Ag/GO–grGO/Au) generates a voltage of the highest value under the synergistic effect of three parts. Moreover, a series of control experiments including environmental RH variation, sample thickness, device size, and laser irradiation time further confirm the proposed mechanism (Supplementary Figs 12–14).

**Demonstration of application.** Additionally, the output of the devices is further scaled up by simple series and parallel

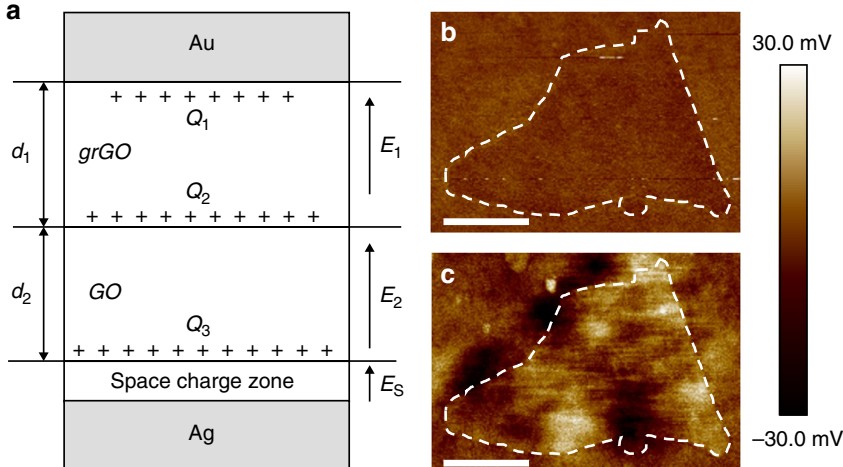

**Fig. 5** Mechanism for the hygroelectric generator. **a** Theoretical model for the proposed working principle of HEG (Ag/GO-grGO/Au) in which the Au electrode contacts with graphene oxide (GO) side of heterogeneous graphene oxide (h-GO) while Ag electrode contacts with gradient reduced graphene oxide (grGO) side of h-GO. The h-GO is composed of grGO and GO with a thickness of $d_1$ and $d_2$, respectively. $Q_1$, $Q_2$, and $Q_3$ represents the induced positively charge quantity at Au/grGO, GO/grGO, and GO/Ag interfaces, respectively. The built-in electric field at grGO, GO and space charge zone is represented by $E_1$, $E_2$, and $E_s$, respectively. **b**, **c** Electrostatic force microscopy (EFM) images of graphene oxide (**b**) and in situ laser-induced reduced graphene oxide (**c**) sheets at low relative humidity. The color bar represents the surface potential variation. Scale bars: 1 μm

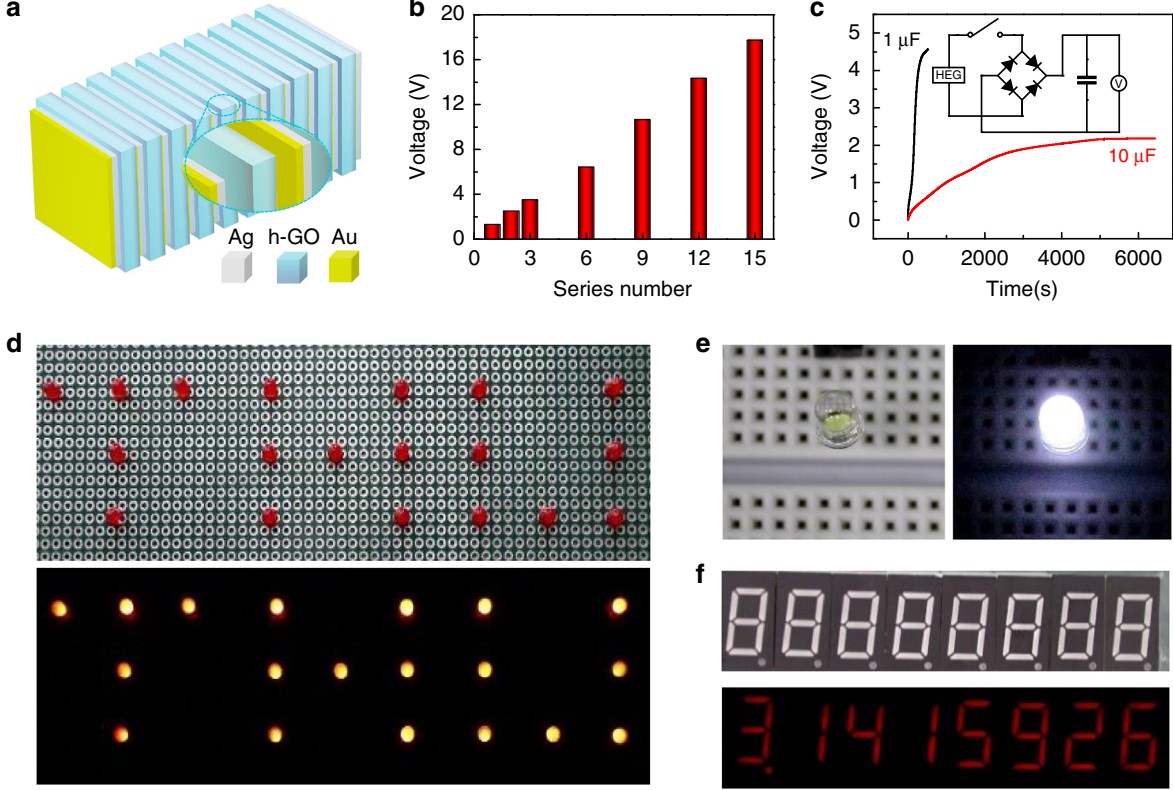

**Fig. 6** Demonstration of hygroelectric generator as practical power source. **a** Schematic of an assembled hygroelectric generator (HEG) package in which Ag and Au electrodes are used to construct Schottky and Ohmic contacts at the two ends of the heterogenous graphene oxide (h-GO), respectively. **b** Voltage output of HEG package with incremental device units in series. **c** Voltage–time curves of commercial capacitors charging by HEG package with 12 device units. The inset is the charge storage circuit. **d**–**f** Various commercial electronics powered by the stored charge in 1 μF capacitor (**c**), such as (**d**) a "THU" pattern constructed by 19 red light-emitting diodes, (**e**) white light-emitting diode with a working potential of 2.6 V, and (**f**) eight-bit digital display tube showing "3.1415926"

connections of HEG (Ag/GO–grGO/Au). To realize a satisfying scaling performance on HEGs, the concurrent RH variation and simultaneously electrical response of each unit are significantly vital. For this purpose, we herein develop a rather facial and highly efficient stacking method for devices integration. Benefiting from the sandwiched structure, hygroscopic h-GO of each unit can easily adsorb water molecules from the environment at almost the same time due to the fully exposed active sites and

superb water uptake ability (Supplementary Fig. 15), leading to the synchronously charge separation and proton migration within different units. As shown in Fig. 6a, 15 HEGs power units with layer-by-layer stacking strategy show remarkable 18 V output (Fig. 6b). To demonstrate the practical application of the HEG, a designed system is developed by integrating a power package containing ten HEGs units with an alternating current-to-direct current converting circuit and a commercial capacitor as shown in Fig. 6c. The capacitor can be charged up to 4.5 V (1 μF) within 500 s or 2 V (10 μF) within 5000 s once exposing the HEG package to moisture. It is sufficient to power a series of commercial electronics, such as a "THU" pattern containing 19 red light-emitting diodes (LEDs) in series (Fig. 6d and Supplementary Movie 1), commercial white LED with a working voltage of 2.6 V (Fig. 6e and Supplementary Fig. 16), and nixie tube arrays showing "3.1415926" (Fig. 6f).

## Discussion

In summary, we report a high-performance HEG based on h-GO bulk via directionally controlled laser irradiation method. The HEG can generate remarkably high-voltage output that approaches 1.5 V, which is the highest value among HEGs reported previously to the best of our knowledge. The electric output performance arises from the synergetic effect of a heterogeneous distribution of oxygen-containing functional groups and a well-matched interface system. Furthermore, the HEG can be easily integrated with superior consistency using a simple and effective device stacking strategy, which delivers remarkable electric energy for powering a series of electronics including LED arrays and digital display tubes. With this performance, we expect that the designed HEG could be applied not only as a portable power source for self-powered electronics, but also possibly in self-powered sensors for applications in the Internet of Things.

## Methods

**Preparation of heterogeneous graphene oxide assembly**. GO was synthesized by oxidation of natural graphite powder using modified Hummers' method as reported previously[7]. The porous 3D GO aerogel was prepared by freeze-drying of as-synthesized GO dispersions (5 mg ml$^{-1}$, 5 ml) in a columnar polyethylene tube with a diameter of 20 mm for 3 days. Further tableting process was conducted to enhance the mechanical properties of bulk GO under a press of 10 kN for 10 min. The heterogeneous GO assembly was prepared via a 450 nm laser (15 W) with a 5 cm$^2$ area at ambient condition.

**Electric measurements**. The h-GO assembly was sandwiched by a pair of Au and Ag electrodes and inserted to a test circuit in an enclosed container with a RH controlling system. The circuit parameters of the open-circuit voltage test were current = 0 nA and step index = 10 points s$^{-1}$. The circuit parameters of the short-circuit current test were voltage = 0 mV and step index = 10 points s$^{-1}$.

**Characterization**. The morphology and composition of as-prepared samples were investigated by scanning electron microscope (SEM, FLexSEM 1000) and energy-dispersive analysis of X-rays (EDAX, AZtec). X-ray diffraction (XRD) patterns were recorded on a Bruker AXS D2 PHASER diffractometer with a Cu Kα irradiation source (λ = 1.54 Å). X-ray photoelectron spectra were performed by a PHI Quantera II (Ulvac-Phi Incorporation) photoelectron spectrometer with Al Kα (1846.6 eV). Raman spectra were carried out on a LabRAM HR Raman spectrometer (Horiba Jobin Yvon) with a 532 nm laser. Fourier-transform infrared (FT-IR) spectra were collected by UATR Two FT-IR spectrometer. Atomic force microscopic (AFM) and electrostatic force microscopic (EFM) images were taken using an Innova (Bruker) atomic force microscope. The voltage and current signals were recorded in real time using a Keithley 2612 multimeter, which was controlled by a LabView-based data acquisition system.

## Data availability

The data that support the findings of this study are available from the authors upon reasonable request.

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

## Acknowledgements

We acknowledge the financial support from the National Key R&D Program of China (2017YFB1104300, 2016YFA0200200), NSFC (No. 51673026, 51433005), NSFC-MAECI (51861135202), and Beijing Municipal Science and Technology Commission (Z161100002116022).

## Author contributions

Y.H., H.C., and L.Q. contributed to the project design. H.C. and L.Q. elaborated the methodology and supervised the study. Y.H. conceived and carried out the majority of experimental measurements. C.Y., P.Z., Q.L., and H.Y. helped with the fabrication and characterization of materials and the devices. Y.H., H.C., G.S., and L.Q. analyzed the data and discussed the results. Y.H., H.C., and L.Q. wrote the paper.

## Additional information

**Competing interests:** The authors declare no competing interests.

