## [Peer Review File · Nature Communications]

Reviewers' comments:

Reviewer #1 (Remarks to the Author):

This manuscript reports a hygroelectric generator (HEG) unit by hygroscopic GO bulk and interface mediation. A single hygroelectric generator can reach a record high output voltage of 1.5 V. The research is interesting and the data are convincing. Therefore, I will recommend acceptance at Nature Communications. Below are some suggestions for the authors to further enhance the manuscript quality.

1. Whether the increase of electrical resistance with the introduction of GO will affect the performance of the devices? More explanations will be very helpful.
2. The influence of thickness on the performance of the devices may be better discussed.
3. Whether the size of the hygroscopic GO bulk will affect the performance of the devices? With the increase of size, the adsorption amount of water may increase while the resistance of the devices may also increase. How will these influence the generated voltage and current?
4. In the description of Fig. 6, the unit of capacitance needs to be carefully checked. There was μF in some sentences while in others, uf was used. So does the corresponding description in the main body.
5. References need to be carefully checked as some information may be updated. For examples, the information of ref.28 He, S., et al. Chemical-to-electricity carbon: water device. Adv. Mater. 0, 1707635 (2018) has already been updated.
6. Some sentences in this manuscript seem to be confusing to readers. For example, "The huge electric output performance was originated from the synergetic effect heterogeneous distribution of oxygen containing functional groups and well-matched interface system." There seem to be some grammar mistakes in this sentence.

Reviewer #2 (Remarks to the Author):

In this work, the authors demonstrate a high-performance hygroelectric generator based on porous heterogeneous GO foam with outputting approaching to 1.5V open voltage of a single power unit when experienced 80% RH variation, which is comparable to a commercial dry cell. They attribute the excellent performance to the heterogeneous reconfiguration of oxygen-containing functional groups on hygroscopic GO and the mediation on electrodes/functional materials interfaces. A series of systematic and careful control experiments have been done regarding the functional materials and device configuration and a theoretical analysis has been established based on a plate capacitance model. The experimental results are well consistent with the proposed models. Moreover, the authors successfully scaled up the devices with outputting remarkably high voltage, i.e., 18V with 15 power units, and they also successfully used the generated electricity to power various electric devices. The generated electric voltage is of practical importance and is rather higher than other water-related electric generators.

The results are novel and appealing, and will be of great interest for researchers working in the energy- and environment- related fields, and for a broader community. The work is well organized at the high professional level of Nature Communications and should be published after addressing the minor questions.

1. In the introduction section, the authors talk about many GO based HEGs in responding to RH change. In the water-enabled energy-harvesting field, there are still many other forms to generate electricity, i.e., evaporation-induced electric energy. Authors should introduce comprehensive insights regarding to this area with parallel comparisons.
2. In fig 6a. Authors claim the HEGs can be easily scaled up by a stacking method. The electric performance of HEGs is significantly dependent on the different electrodes and a suitable device configuration. How to assure the consistency of the device in the integrated state.
3. The authors should modify the graphs for clear especially the axis and the font size of fig. 1c, d, fig. 4 and fig. 6b, c.
4. The authors should carefully recheck the spelling mistakes through the manuscript.

Reviewer #3 (Remarks to the Author):

In this manuscript, the authors have developed a novelty hygroelectric generator (HEG) that can output a high voltage of about 1.5 V triggered by humidity variation. The brand-new and well-deigned synergy of HEG is impressive, which is benefited from the developed heterogeneous GO bulk and interface mediation between electrodes/materials with Schottky junctions. In addition, the authors have explored an easily serial connection strategy on HEGs, scaling up the final voltage to be a remarkable value of 18 V and powering many commercial electronic devices. This study about HEG is an innovative research, which will potentially cater to broad readership in the field of energy conversion from moisture change to electricity and their further applications.

This manuscript has carefully studied the construction of HEG, the preparation of inner heterogeneous GO bulk and the effect of interface mediation. The moisture triggered electricity generation process is also elaborated by experimental data. It demonstrated a practical application of HEG to indeed take use of environmental energy where widely existed in natural world. The description is clear and the underlying mechanism is well clarified and supported by experimental results combined with theoretical analysis. In this regard, this work is certainly qualified for the publication in Nature Communications. I would recommend its acceptance of this good work in Nature Communications. Besides this already overall work in the manuscript, what interested me is whether this electricity generation of HEG could occur under other conditions like thermal variation? If so, it would further expand the applications of this new HEG in practical use at various places and be helpful to the further development of the field in future.

Reviewers' comments:

Reviewer #1 (Remarks to the Author):

This manuscript reports a hygroelectric generator (HEG) unit by hygroscopic GO bulk and interface mediation. A single hygroelectric generator can reach a record high output voltage of 1.5 V. The research is interesting and the data are convincing. Therefore, I will recommend acceptance at Nature Communications. Below are some suggestions for the authors to further enhance the manuscript quality.

Response: We really appreciate the referee for recommending acceptance of our manuscript.

1. Whether the increase of electrical resistance with the introduction of GO will affect the performance of the devices? More explanations will be very helpful.

Response: Thanks for the good comments. We have investigated the correlation between the electric performance of HEG and the electrical resistance of heterogeneous GO (h-GO). The electrical resistance of h-GO is adjusted by controlling the laser irradiation time in the preparation process that mainly causes the change of oxygen containing functional groups (Fig. R1a). As shown in Fig. R1b, the voltage output of HEG has minor change while the current reduces observably with the increased electrical resistance of h-GO (from 5.3 MΩ to 9.2 MΩ).

In the electricity generation process, spontaneous absorption and subsequent hydration in hygroscopic h-GO result in charges separation of positive-negative ions pair and subsequently release positively charged protons by ionization effect when exposed to water molecules. Then, the positively charged protons would diffuse from high-concentration side (GO layer) to low-concentration side (grGO layer) under the drive of gradient structure, which induces a current flow and an electric potential difference between two electrodes, thus outputting electric power. The induced voltage of HEG is mainly attributed to the charges partition of positive-negative ions pair and the induced current is ascribed to the migration of positively charged protons within h-GO. As indicated in equation (4) in main text as below:

$$V = \left(\frac{(Q_2 - Q_1)d_1}{s\varepsilon_0\varepsilon_1} + \frac{(Q_3 - Q_2)d_2}{s\varepsilon_0\varepsilon_2} + V_s \right) \quad (R1)$$

The generated voltage (V) is proportional to the difference of charge quantity of $(Q_2 - Q_1)$ and $(Q_3 - Q_2)$. The reduced oxygen containing functional groups on h-GO along with increased laser irradiation time would induce the electrical resistance decrease of h-GO, thus resulting depletion in numbers of positively charged protons and changes of Q_1 , Q_2 and Q_3 . However, this concurrent reduction on Q_1 , Q_2 and Q_3 would bring insignificant change on the difference of charge quantity $(Q_2 - Q_1)$ and $(Q_3 - Q_2)$ in equation (R1). Therefore, the generated voltage of HEG exhibits negligible changes. However, the positively charged protons diffusion is severely hindered by the increased electrical resistance of h-GO, leading to observably weakened current output of HEG.

Fig. R1 (a) Electrical resistance of h-GO and C/O atomic ratio at the grGO surface of h-GO upon different laser irradiation time. The electrical resistance of h-GO notably decreases from 9.2 MΩ to 5.3 MΩ with prolonging the laser irradiation time from 1 h to 8 h, accompanying by an increase of C/O atomic ratio from 2.3 to 3.05 at the grGO surface of h-GO. (b) Voltage, current output in response to the $\Delta RH = 80\%$ of HEG (Ag/GO-grGO/Au) with different electrical resistance of h-GO.

2. The influence of thickness on the performance of the devices may be better discussed.

Response: We are grateful to the reviewer's good comments. The influence of thickness of h-GO on the performance has been studied in the revised manuscript. As shown in Fig. R2a, the induced voltage is enhanced with the thickness increase of h-GO, while the induced current is weakened accordingly. As we have demonstrated above (**Comments 1**), the electricity generation of HEG is significantly dominated by the migration of protons and the charges separation process in h-GO. The generated voltage of HEG mainly depends on the charges partition of positive-negative ions pair within h-GO and can be represented in Equation (R1), where d_1 and d_2 represent the thickness of grGO layer and GO layer within h-GO, respectively. This means the generated voltage (V) is proportional to the thickness of h-GO (d_1 and d_2). Therefore, the generated voltage is enhanced with incremental thickness of h-GO. However, the increased thickness of h-GO could prolong the water absorption time and increase the migration path for protons that can gradually reduce the ionic conductivity of h-GO, which should be the reason that weakens the output current value of HEG.

To further confirm the ionic conductivity change of h-GO with increased thickness of h-GO, the electrochemical impedance spectroscopy measurements are implemented on the h-GO with different thickness. For the ionic conductivity measurement, the impedance spectra obtained yielded a depressed semicircle with a slanted line at lower frequencies as shown in Fig. R2b. In solid electrolyte system, the corresponding equivalent circuit for this type of spectra is typically represented by electrode resistance (R_s) in series with a parallel combination of electrolyte resistance (R_e) and capacitance^{R1, R2}. Therefore, the impedance data at high frequencies are fitted by Z-view based on the equivalent circuit (inset Fig. R2b), where the depressed semicircles are simulated by the electrolyte resistance in parallel with a Constant

Phase Element (CPE) that is generally a result of electrode roughness. The experimental results show that the ionic resistance is actually increased with the thickness of h-GO, indicating a poor ionic conductivity for thicker samples. Therefore, the induced current is weakened whereas the output voltage is promoted with an increase in thickness of h-GO.

Fig. R2 (a) Voltage, current output in response to $\Delta RH = 80\%$ of HEG (Ag/GO-grGO/Au) with different h-GO thicknesses. (b) The electrochemical impedance spectra of h-GO with different thickness (RH~ 90%, 25 °C).

3. Whether the size of the hygroscopic GO bulk will affect the performance of the devices? With the increase of size, the adsorption amount of water may increase while the resistance of the devices may also increase. How will these influence the generated voltage and current?

Response: We strongly appreciate reviewer's valuable and helpful comments. We have carried out additional experiments to evaluate the influence of area size on the device performance. According to the law of resistance, the electrical resistance (R) of h-GO is determined by

$$R = \rho \frac{L}{A} \quad (R2)$$

Where ρ , L and A represents the resistivity, thickness and area size of h-GO, respectively. Thus, the electrical resistance of h-GO reduces with increased area size, which is consistent with the experimental results (Fig. R3a). The electricity generation of HEG presents a positive dependency between the area size and the output current, coupling with a slightly weakened output voltage with increased area size of h-GO (Fig. R3b). This phenomenon could be ascribed to the enhancement effect of improved hydration based on size increase. As shown in Fig. R3c, the water uptake ability of h-GO is remarkably enhanced with incremental area size, which could causes more ionized protons and facilitates its migration within h-GO^{R2}, giving rise to enhanced output current. The slightly shrinking output voltage with increased size of h-GO could be attributed to the uneven absorption of water molecules due to the incremental sample size.

Fig. R3 (a) Resistance variation of h-GO samples with different area size under specific humidity. (b) Voltage, current output in response to the $\Delta RH = 80\%$ of HEG (Ag/GO-grGO/Au) with an h-GO thickness of 100 μm and different area size. (c) The water absorption characteristic of samples with different area size.

4. In the description of Fig. 6, the unit of capacitance needs to be carefully checked. There was μF in some sentences while in others, uf was used. So does the corresponding description in the main body.

Response: We have carefully checked the unit of capacitance throughout the manuscript and modified to correct way in the revised manuscript.

5. References need to be carefully checked as some information may be updated. For examples, the information of ref.28 He, S., et al. Chemical \square to \square electricity carbon: water device. Adv. Mater. 0, 1707635 (2018) has already been updated.

Response: Thank you for your good comments. We have updated the references accordingly.

6. Some sentences in this manuscript seem to be confusing to readers. For example, "The huge electric output performance was originated from the synergetic effect heterogeneous distribution of oxygen containing functional groups and well-matched interface system." There seem to be some grammar mistakes in this sentence.

Response: We truly thank for the kind suggestions from the referee. We have carefully examined the full manuscript and corrected the grammatical and clerical errors in the revised manuscript and Supplementary Information.

Reviewer #2 (Remarks to the Author):

In this work, the authors demonstrate a high-performance hygroelectric generator based on porous heterogeneous GO foam with outputting approaching to 1.5V open voltage of a single power unit when experienced 80% RH variation, which is

comparable to a commercial dry cell. They attribute the excellent performance to the heterogeneous reconfiguration of oxygen-containing functional groups on hygroscopic GO and the mediation on electrodes/functional materials interfaces. A series of systematic and careful control experiments have been done regarding the functional materials and device configuration and a theoretical analysis has been established based on a plate capacitance model. The experimental results are well consistent with the proposed models. Moreover, the authors successfully scaled up the devices with outputting remarkably high voltage, i.e., 18V with 15 power units, and they also successfully used the generated electricity to power various electric devices. The generated electric voltage is of practical importance and is rather higher than other water-related electric generators.

The results are novel and appealing, and will be of great interest for researchers working in the energy- and environment- related fields, and for a broader community. The work is well organized at the high professional level of Nature Communications and should be published after addressing the minor questions.

Response: We greatly acknowledge the referee for the positive comments on our work.

1. In the introduction section, the authors talk about many GO based HEGs in responding to RH change. In the water-enabled energy-harvesting field, there are still many other forms to generate electricity, i.e., evaporation-induced electric energy. Authors should introduce comprehensive insights regarding to this area with parallel comparisons.

Response: Thanks for the good suggestions. As the referee suggested, we have briefly summarized previous researches about water-enabled energy-harvesting systems and added them into the introduction part in the revised manuscript.

2. In fig 6a. Authors claim the HEGs can be easily scaled up by a stacking method. The electric performance of HEGs is significantly dependent on the different electrodes and a suitable device configuration. How to assure the consistency of the device in the integrated state.

Response: We greatly thank the reviewer's valuable comments. Pertaining to the electricity generation process of the developed HEG in present work, the performance of HEG is significantly relied on the variation of RH. Considering the scaling process, the concurrent stimuli of each unit plays an important role for devices integration. In this way, the consistency of RH change for many devices is vital. Benefited from the sandwiched structure of HEG and stacked configuration of final serial device, the margins of inner h-GO could be fully exposed to ambient environment. Once exposed to moisture, hygroscopic h-GO of each unit can easily adsorb water molecules from environment at the same time^{R3}. Therefore, this makes sure the synchronously charges separation and protons migration within different units, thus leading to the consistency of electricity generation.

Fig. R4 Schematic illustration of stacking HEGs integration with concurrent moisture stimuli and synchronously protons migration.

3. The authors should modify the graphs for clear especially the axis and the font size of fig. 1c, d, fig. 4 and fig. 6b, c.

Response: We greatly thank the referee for these constructive suggestions. We have modified the figures accordingly.

4. The authors should carefully recheck the spelling mistakes through the manuscript.

Response: Thanks for the kind suggestions from the referee. We have carefully examined the full manuscript and corrected some grammatical and clerical errors in the revised manuscript and Supplementary Information.

Reviewer #3 (Remarks to the Author):

In this manuscript, the authors have developed a novelty hygroelectric generator (HEG) that can output a high voltage of about 1.5 V triggered by humidity variation. The brand-new and well-deigned synergy of HEG is impressive, which is benefited from the developed heterogeneous GO bulk and interface mediation between electrodes/materials with Schottky junctions. In addition, the authors have explored an easily serial connection strategy on HEGs, scaling up the final voltage to be a remarkable value of 18 V and powering many commercial electronic devices. This study about HEG is an innovative research, which will potentially cater to broad readership in the field of energy conversion from moisture change to electricity and their further applications.

This manuscript has carefully studied the construction of HEG, the preparation of inner heterogeneous GO bulk and the effect of interface mediation. The moisture triggered electricity generation process is also elaborated by experimental data. It demonstrated a practical application of HEG to indeed take use of environmental energy where widely existed in natural world. The description is clear and the underlying mechanism is well clarified and supported by experimental results

combined with theoretical analysis. In this regard, this work is certainly qualified for the publication in Nature Communications. I would recommend its acceptance of this good work in Nature Communications.

Response: We sincerely thank the referee for appreciating and recommending our work for publication in Nature Communications.

Besides this already overall work in the manuscript, what interested me is whether this electricity generation of HEG could occur under other conditions like thermal variation? If so, it would further expand the applications of this new HEG in practical use at various places and be helpful to the further development of the field in future.

Response: We greatly thank the referee for bringing this new insight to us. As we know, thermal difference induced thermoelectric generators have aroused tremendous interest to harvest electric energy from nature or waste heat in industrial production^{R4},^{R5}. As for the present HEG, thermal difference could be another approach to affect the RH variation and then induce the electricity generation, which is an attractive aspect for our present work. This inspires us to consider the possibility of electricity generation on HEG by thermal difference in our future work.

References

- R1. Zhao, F., Liang, Y., Cheng, H. H., Jiang, L., Qu, L. T. Highly efficient moisture-enabled electricity generation from graphene oxide frameworks. *Energy Environ. Sci.* **9**, 912-916 (2016).
- R2. Gao, W., *et al.* Direct laser writing of micro-supercapacitors on hydrated graphite oxide films. *Nat. Nanotechnol.* **6**, 496-500 (2011).
- R3. Bi, H. C., *et al.* Ultrahigh humidity sensitivity of graphene oxide. *Sci. Rep.* **3**, 7 (2013).
- R4. Bell, L. E. Cooling, heating, generating power, and recovering waste heat with thermoelectric systems. *Science* **321**, 1457-1461 (2008).
- R5. Snyder, G. J., Toberer, E. S. Complex thermoelectric materials. *Nat. Mater.* **7**, 105-114 (2008).

REVIEWERS' COMMENTS:

Reviewer #1 (Remarks to the Author):

The authors have addressed my concerns, and it can be accepted now.

REVIEWERS' COMMENTS:

Reviewer #1 (Remarks to the Author):

The authors have addressed my concerns, and it can be accepted now.

Response: We thank the reviewer very much for the positive comments and kind recommendation.